# Coupling Plasmonic and Cocatalyst Nanoparticles on N–TiO_2_ for Visible-Light-Driven Catalytic Organic Synthesis

**DOI:** 10.3390/nano9030391

**Published:** 2019-03-07

**Authors:** Yannan Wang, Yu Chen, Qidong Hou, Meiting Ju, Weizun Li

**Affiliations:** College of Environmental Science and Engineering, Nankai University, Tianjin 300350, China; wangyannan@nankai.edu.cn (Y.W.); chenyu0870@gmail.com (Y.C.); houqidong@nankai.edu.cn (Q.H.); jumeit@nankai.edu.cn (M.J.)

**Keywords:** plasmonic photocatalyst, metal nanoparticle, N–TiO_2_, nanocomposites, photocatalytic selective oxidation

## Abstract

The use of the surface plasmon resonance (SPR) effect of plasmonic metal nanocomposites to promote photocarrier generation is a strongly emerging field for improving the catalytic performance under visible-light irradiation. In this study, a novel plasmonic photocatalyst, AuPt/N–TiO_2_, was prepared via a photo-deposition–calcination technique. The Au nanoparticles (NPs) were used herein to harvest visible-light energy via the SPR effect, and Pt NPs were employed as a cocatalyst for trapping the energetic electrons from the semiconductor, leading to a high solar-energy conversion efficiency. The Au_2_Pt_2_/N–TiO_2_ catalyst, herein with the irradiation wavelength in the range 460–800 nm, exhibited a reaction rate ~24 times greater than that of TiO_2_, and the apparent quantum yield at 500 nm reached 5.86%, indicative of the successful functionalization of N–TiO_2_ by the integration of Au plasmonic NPs and the Pt cocatalyst. Also, we investigated the effects of two parameters, light source intensity and wavelength, in photocatalytic reactions. It is indicated that the as-prepared AuPt/N–TiO_2_ photocatalyst can cause selective oxidation of benzyl alcohol under visible-light irradiation with a markedly enhanced selectivity and yield.

## 1. Introduction

Titanium dioxide (TiO_2_) was extensively studied in the past two decades as a photocatalyst because it can eliminate environmental pollutants, purify air, and produce clean hydrogen energy through the efficient utilization of solar energy [1]. Nevertheless, owing to the rapid recombination rate of the photogenerated electron–hole pairs and limited visible-light response, the application of pure TiO_2_ is restricted. Appropriate modification, such as doping non-metals, is essential for TiO_2_ to allow the further utilization of solar energy [2]. Nonetheless, the reported reactivity and quantum efficiency of TiO_2_ derivative materials remains extremely low to meet the requirements of practical applications.

In order to sufficiently improve photocatalytic efficiency, both the visible-light absorption region and electron–hole separation of the photocatalyst should be optimized. Recently, semiconductor nanomaterials decorated with noble-metal nanoparticles (NPs) were recognized as a promising method for boosting the performance of photocatalysts [3,4,5,6,7,8]. Coupling semiconductors with noble metals (such as platinum and palladium) as the cocatalyst can form a Schottky barrier, serving as the “electron trapper” to improve charge migration and separation [9,10]. Plasmonic metals (gold and silver) nanoparticles with attractive SPR properties under visible-light excitation can be used as antennas for converting light energy into a local electric field [7,11,12], and improve the photocarrier generation/separation via plasmon-induced resonance energy transfer (PIRET) and the hot-electron injection mechanism [13,14]. Consequently, the combination of multi-functional metal NPs in a noble metal/semiconductor nanostructure might effectively enhance the generation of photo-carriers and strengthen charge migration and separation.

For the hot-electron injection effect, the so-called SPR-sensitization effect, the plasmonic metal nanoparticles act as a dye molecule in dye-sensitized solar cells; as excited by the incident high-energy photon, the SPR effect of the plasmonic metal causes confined free electrons oscillating with incident light to generate excitation of hot electrons via non-radiative decay, so-called “plasmonic hot-electron generation”. Furthermore, these hot electrons with energy high enough to overcome the Schottky barrier can inject into the adjacent semiconductor’s conduction band [15,16,17]. To facilitate the PIRET process, the existence of intra-bandgap level-related defects can play a crucial role in the promotion of PIRET [18,19]. For instance, a TiO_2_ photoanode based on N-doped exhibits enhanced photocurrent behavior induced by a PIRET water-splitting reaction [20]. N-doped TiO_2_ introduces a new intra-bandgap level, which causes the absorption range of the semiconductor photocatalyst to overlap with the extinction wavelength of the plasmonic material, thereby obtaining a sufficient resonance interaction. In this case, the energy of the plasmonic oscillation is transferred from the plasmonic material to the semiconductor photocatalyst by an electromagnetic field or a dipole–dipole interaction. 

In order to fully understand the excellent photocatalytic activity of the bifunctional noble-metal-modified N-doped TiO_2_ under visible-light excitation, a detailed comparative study of light intensity and light wavelength is required. Herein, we integrated the plasmonic effect and a Schottky junction into one nanostructure by forming bifunctional plasmonic photocatalyst co-decorated with Au and Pt NPs and N–TiO_2_. The activities of the AuPt/N–TiO_2_ samples were evaluated by photocatalytic oxidation of benzyl alcohol. By strictly limiting the effects of other factors, we observed a direct correlation between photocatalytic activity and the irradiation parameter, which is essential for the design to improve the efficiency of the photocatalytic reaction. The results obtained in this paper are expected to contribute to the rational design and development of multifunctional metal nanoparticles for applications targeting solar energy conversion.

## 2. Materials and Methods 

### 2.1. Synthesis

In a typical procedure, TiO_2_-supported nanocrystals were prepared according to our previous paper [21]. The prepared TiO_2_ product was mixed and ground with urea (1:4), and then the mixture was heated in air at 400 °C for two hours to obtain N–TiO_2_ [22]. Finally, the noble metals were deposited on the N–TiO_2_ via a typical photo-deposition calcination method, and the as-prepared catalysts were annealed in air for further use. All experimental methods are fully reported in the Appendix A.

### 2.2. Sample Characterization

X-ray diffraction (XRD) patterns of the samples were recorded using a PANalytical X’pert MRD system (Almelo, Netherlands). Diffuse-reflectance ultraviolet–visible light (UV–Vis) spectra (DRS) of the samples were recorded on a Shimadzu 3600 UV–Vis spectrophotometer (Kyoto, Japan) in the air against BaSO_4_. Transmission electron microscopy (TEM) images and scanning electron microscopy (SEM) images were taken by a FEI Tecnai F20 microscope (Hillsboro, OR, USA) and Hitachi S4800 microscope (Tokyo, Japan). X-ray photoelectron spectra (XPS) of the samples were recorded on a Thermo-Fisher Scientific ESCALAB 250XI system (Waltham, OR, USA). The steady-state photoluminescence (PL) spectrum was recorded by a Hitachi F-7000 fluorescence spectrophotometer (Tokyo, Japan). Photocurrent and electrochemical impedance spectroscopy measurements of the photocatalyst were performed on a CHI 760D workstation (Shanghai, China). All electrochemical measurements were made at room temperature. 

## 3. Results and Discussion

SEM and TEM images (Figure 1 and Appendix A) were recorded to observe the morphology of as-prepared Au_2_Pt_2_/N–TiO_2_. As shown in the TEM images, gold nanoparticles with an average size of ~25 nm can be observed instead of the presence of bimetallic Au–Pt alloy. It is suggested that the size of the metal nanoparticles is critical for modifying the chemical composition of the resulting nanomaterials, and the bimetallic alloy usually can be observed in the case of small gold (~7 nm) and platinum (~2 nm) nanoparticles [23]. Here, in this case, the gold nanoparticle size was larger than 20 nm, the formation of the bimetallic alloy NPs was not thermodynamically favored, and the segregation of gold and platinum nanoparticles was maintained. On the other hand, the platinum metal NPs was observed with a mean size of 2 nm (Figure 1h), which was uniformly decorated on the N–TiO_2_ support. Furthermore, interplanar distances of 0.235 and 0.225 nm for gold and platinum NPs were observed in the high-resolution TEM images (Figure 1f,g), which were indexed to the lattice spacings of Au(111) and Pt(111) planes of face-centered cubic (fcc) structures, respectively [24]. Another type of lattice fringe (~0.352 nm) can be indexed to the (101) plane of anatase TiO_2_. The results obtained from the energy-dispersive X-ray (EDX) spectrum showed that the noble metal’s composition in Au_2_Pt_2_/N–TiO_2_ (list in Appendix A) was consistent with the nominal load. Furthermore, the Brunauer–Emmett–Teller (BET) characterization results (Appendix A) indicated that all catalysts exhibited similar surface areas, while pore volume and pore size decreased after the loading of noble-metal nanoparticles.

In the XRD characterization, all catalysts exhibited diffraction peaks dominated by TiO_2_ (Figure 2a). Concerning the Au NPs, some additional weak peaks observed at 38° corresponded to Au; however, diffraction peaks for Pt were not found in the XRD patterns, possibly related to the line broadening caused by the quantum-size effects of small-sized Pt NPs [25]. Figure 2b shows the UV–Vis absorption spectra (DRS) of the as-prepared catalyst. The absorption band of the N–TiO_2_ sample in the visible region of 400–500 nm corresponds to the presence of nitrogen. This effect is related to nitrogen doping, which possibly leads to the formation of hybridized states at the top of the valence band of the nitrogen *2p* states and oxygen *2p* states or an N-induced intermediate gap level [26]. XPS profiles were recorded to investigate the localization of nitrogen. N was mainly located at the interstitial atom on the Ti–O–N bonds (Appendix A). By introducing impurity levels into the TiO_2_ lattice, more overlapping portions of the absorption spectrum can be obtained between the TiO_2_ and Au nanoparticles. In this way, the near-field electromagnetic resonance of the SPR effect can collect enough energy to stimulate the generation of electron–hole pairs through the PIRET process. Compared to N–TiO_2_, Au_2_Pt_2_/N–TiO_2_ exhibited a stronger absorption feature around 550 nm in Figure 2b, corresponding to the SPR peak of Au NPs [27]. As reported previously [6,28,29,30,31], plasmonic nanoparticles (gold, silver) loaded on a semiconductor, with a broad absorption cross-section, are capable of absorbing visible light and generating hot electrons through intraband transitions. These high-energy hot electrons then overcome the Schottky barrier and inject into the conduction band of the semiconductor. In this way, the SPR effect of the metal nanoparticles leads the photon energy transfer to the adjacent semiconductor or molecular complex, which in turn drives the chemical reaction. 

Also, the photoluminescence emission (PL) of the samples was recorded to understand the behavior of the electrons and holes generated by light in catalysts. Here, the steady-state fluorescence emission spectrum (Figure 2c) showed a substantial attenuation of the PL signal owing to the deposition of noble metal (Pt), indicating that Pt NPs effectively form the Schottky barrier at the metal/N–TiO_2_ heterojunction. This Schottky barrier, in turn, reduces electron–hole (e^−^–h^+^) pair recombination and increases the number of photoreactive photo-carriers available for photoreaction [32]. To further determine the role of the noble-metal nanoparticles in illumination, the photoelectrochemical properties of catalyst were characterized (Figure 2d), and it is demonstrated that the photocurrent intensity of Au_2_Pt_2_/N–TiO_2_ is considerably higher than that of N–TiO_2_. Such an apparent transient photocurrent enhancement is primarily associated with the available gold NPs, which absorb visible light and promote photocarrier generation through the SPR effect. Subsequently, we used the electrochemical impedance spectra (EIS) experiments to investigate the generation of the electron. The results of charge transport characteristics (Figure 2d inset) revealed that the radius of Au_2_Pt_2_/N–TiO_2_ in the middle-frequency region is smaller than the radius of N–TiO_2_, which demonstrates that the photoinduced electron–hole separation efficiency is higher, and the interface charge can be transferred to the electron donor more quickly. 

For investigating the photocatalytic performance, we used the selective oxidation of benzyl alcohol as a probe reaction to study the photocatalytic activity of Au_2_Pt_2_/N–TiO_2_ catalyst for visible-light-driven organic catalytic synthesis [33,34,35]. Figure 3a summarizes the reaction parameters such as conversion, yield, and selectivity data. After 2.5 h, benzaldehyde formed over bare TiO_2_ (yield: 3.37%) under visible-light irradiation. Considering that bare TiO_2_ does not absorb visible light, this visible-light catalytic reactivity can be ascribed to the ligand-to-metal charge transfer resulting from the surface complex formed by the adsorption of benzyl alcohol on the surface of TiO_2_ [36,37,38]. Moreover, N–TiO_2_ does not significantly increase the activity, and the selectivity in TiO_2_ and N–TiO_2_ cases was low (~70%). Hence, the loading of metal NPs can significantly improve reaction efficiency compared with TiO_2_. Among all samples, the Au_2_Pt_2_/N–TiO_2_ composite presented the highest photocatalytic performance, and its yield was 24 times that of TiO_2_. On the contrary, the yield of the Au_2_Pt_2_/TiO_2_ photocatalyst was only 70% of the Au_2_Pt_2_/N–TiO_2_, which suggests that the overlapped intrinsic absorption of N–TiO_2_ with plasmonic material may boost the PIRET process. Moreover, the yields over Au_2_/N–TiO_2_ and Pt_2_/N–TiO_2_ increased by 5.5 and 19 times, respectively, indicating that the Schottky barrier formed between Pt nanoparticles and TiO_2_ is crucial for the improvement in the catalyst efficiency. It is interesting to note that, after the loading of noble-metal NPs, the selectivity increased from 73.8% to greater than 95%, which means that, when noble-metal nanoparticles are used as photocatalysts for selective oxidation of benzyl alcohol, the photolysis of the reaction is negligible.

To better understand the factors affecting the performance of Au_2_Pt_2_/N–TiO_2_ photocatalyst, we tuned and investigated the light-source wavelength and intensity in the photocatalytic reaction. The most significant enhancement in the yield of the reaction was observed by irradiation of 460–560 nm over Au_2_Pt_2_/N–TiO_2_ photocatalyst (Figure 3b), accounting for 81.26% of the strengthening of the total light irradiation. Also, we used a multiple-wavelength laser light source to confirm the effect of illumination wavelength (Figure 3d and Appendix A). Au_2_Pt_2_/N–TiO_2_ exhibited an exceptionally high apparent quantum yield at two wavelengths (500 nm and 532 nm), with 5.86% at 500 nm and 4.57% at 532 nm. Moreover, it is believed that there are two possible mechanisms that may affect the performance of the photocatalytic activity, namely hot-electron injection and PIRET. Upon irradiation of visible light, following light absorption and SPR excitation in these nanostructures, electromagnetic decay takes place on a femtosecond timescale non-radiatively by transferring the energy to hot electrons; then, these “hot enough” electrons with high energy would inject into the N–TiO_2_ conduction band. In this manner, the apparent quantum yield will fit well with the pattern of the plasmonic metal absorption spectrum, which is consistent with an observation reported in previous literature [39]. On the other hand, in this case, nitrogen doping introduces a new intra-bandgap level above the TiO_2_ valence band, which can resonate with the electromagnetic field generated by the gold SPR effect, and the electromagnetic field is then able to improve the generation of photocarriers from intra-bandgap levels to the TiO_2_ conduction band through the PIRET process. As a result, it will further increase the photocatalytic efficiency. Therefore, a high apparent quantum yield (AQY) was observed in the band that was contributed by the hot-electron injection mechanism caused by plasmonic absorption and the PIRET mechanism. Subsequently, further analysis (Figure 3c) showed that the correlation between the dependence of light enhancement on optical irradiance of all photocatalysts indicates that photoexcitation intensity is a crucial factor for the photo-enhancing activity, which is consistent with previously published literature [40,41]. The above analysis suggested that the enhanced activity on the Au_2_Pt_2_/N–TiO_2_ catalyst is dominated by the specific illumination wavelength and irradiation intensity. Through the analysis of PL and photoelectrochemical tests, we believe the following two factors can describe this: (1) through the loading with Au NPs (plasmonic nanoparticles), on the one hand, plasmonic photocatalysts can utilize a specific wavelength of photons (especially in the visible-light range) to extract hot electrons from the plasmonic metals and more efficiently generate electron–hole pairs. On the other hand, due to the strong near-field electromagnetic resonance caused by the surface plasmons, the rate of generation of photocarriers in TiO_2_ is enhanced by the PIRET between the electromagnetic field and the resonance electronic energy levels of TiO_2_; (2) the integration of a cocatalyst such as Pt NPs can result in the formation of a Schottky barrier that acts as an “electron trapper” for improving photoinduced charge transport and separation.

The reaction mechanism involved in the photocatalytic oxidation of benzyl alcohol on Au_2_Pt_2_/N–TiO_2_ was inspected by a control experiment using different radical scavengers and by electron spin resonance (ESR) spectroscopy measurement using spin trapping and labeling [42]. As shown in Figure 4a, there was no significant change in the reaction process for the hydroxyl (·OH) radicals scavenged by TBA. However, when ammonium oxalate, silver nitrate, and benzoquinone were separately added to capture photogenerated holes, electrons, and superoxide (·O^2−^) radicals, the yield of the reaction was significantly reduced. This observation indicated that, in addition to the hydroxyl (·OH) radicals, radicals such as photogenerated holes, electrons, and superoxide radicals are involved in the process of visible-light photooxidation of benzyl alcohol. Furthermore, ESR measurement using spin trapping and labeling (Figure 4b) indicated that oxygen can be used to capture photogenerated electrons, providing superoxide ·O^2−^) radicals, which played a vital role in the photocatalytic process. It is a known fact that the ·OH radical is a highly reactive intermediate which can oxidize substrate molecules indiscriminately without selectivity [43]; however, superoxide (·O^2−^) radicals are well-known oxidants for selective oxidation reactions [44,45]. Thus, the specific oxidation behavior of the superoxide species in the system and the absence of hydroxyl (·OH) radicals can advantageously favor the selective oxidation of benzyl alcohol and can be a significant cause of high reaction selectivity. 

The possible reaction mechanism is illustrated in Figure 4c. Under specific wavelength and intensive visible-light irradiation, the incident photons excite the SPR of the gold nanoparticles. The electrons collectively oscillating by localized surface plasmons decay non-radiatively through intraband or interband excitations on Au NPs, generating hot electrons with high enough energy, then finally transfer into the N–TiO_2_ conduction band. Meanwhile, in this case, N-doping introduces a new intra-bandgap level above the TiO_2_ valence band, which can resonate with the electromagnetic field generated by the gold SPR effect, and the electromagnetic field is then able to improve the generation of photocarriers from intra-bandgap levels to the TiO_2_ conduction band through the PIRET process. After the contact of Pt and N–TiO_2_, a Schottky junction can be established at the interface, wherein the conduction and valence band are bent upward to the N–TiO_2_ interface. The electrons on the N–TiO_2_ conduction band are enriched by the Pt NPs via the Schottky barrier between cocatalyst and N–TiO_2_ and then captured by oxygen molecules, affording superoxide (·O^2−^) species [11,46]. The superoxide (·O^2−^) species may attract the hydrogen atom of the substrate (benzyl alcohol) to form an alkoxide intermediate; after that, the transient alkoxide intermediate undergoes rapid hydride transfer, resulting in the elimination of proton hydrogen, and ultimately resulting in benzaldehyde. The local electromagnetic field generated by the gold nanoparticles under visible light enhances the excitation probability of the photogenerated electron and hole pairs of the N–TiO_2_ support material, resulting in more photogenerated carriers, and then these photogenerated electrons migrate to the cocatalyst nanoparticles across the Schottky barrier between Pt–TiO_2_ and react with oxygen to form superoxide radicals.

As listed in Table 1, we further investigated the photocatalytic oxidation of various aromatic alcohols over Au_2_Pt_2_/N–TiO_2_. As expected, Au_2_Pt_2_/N–TiO_2_ has not only high activity for oxidation of aromatic alcohols, but also has excellent selectivity for carbonyl compounds. Furthermore, the conversions of different aromatic alcohol substrates were significantly different; for example, the substitution of *para*-substituted benzyl alcohol with an electron-donating group (–OCH_3_ and –CH_3_) can increase the efficiency of the reaction, while substitution with an electron-withdrawing group (–Cl) lowers the activity. Furthermore, the durability of the photocatalyst is also a crucial parameter for its further application. As shown in Appendix A, no significant decrease in the photocatalytic activity was observed after five cycles, indicating that Au_2_Pt_2_/N–TiO_2_ maintained highly durability in the photocatalytic reaction.

## 4. Conclusions

In this study, we successfully synthesized the plasmonic photocatalyst Au_2_Pt_2_/N–TiO_2_ and investigated its catalytic performance for photo-oxidation of aromatic alcohol. A combination of bifunctional metal NPs was demonstrated for their dual properties related to plasmonic absorption, as well as efficient electron trapping. Coupling a semiconductor with Pt NPs as the cocatalyst can form a Schottky barrier interface, serving as the “electron trapper” to improve charge migration and separation. The Au nanoparticles can be used as an antenna for converting light energy into a local electric field, and hot-electron injection and PIRET mechanisms improve photocarrier generation. The intra-bandgap states of N-doped TiO_2_ take a crucial part in improving both hot-electron injection and PIRET from plasmonic metal nanoparticles to the semiconductor. As a result, the reaction rate of the Au_2_Pt_2_/N–TiO_2_ catalyst, herein, is ~24 times than that of TiO_2_, and the AQY at 500 nm reaches 5.86%, indicative of the successful functionalization of N–TiO_2_ via the integration of Au plasmonic NPs and the Pt cocatalyst. Furthermore, it is indicated that the intensity and wavelength of the illumination source and the choice of the light source have a significant impact on the activity of photocatalytic reaction. This modification of multifunctional metal NPs demonstrates promise for visible-light-driven catalytic applications.

## Figures and Tables

**Figure 1 nanomaterials-09-00391-f001:**
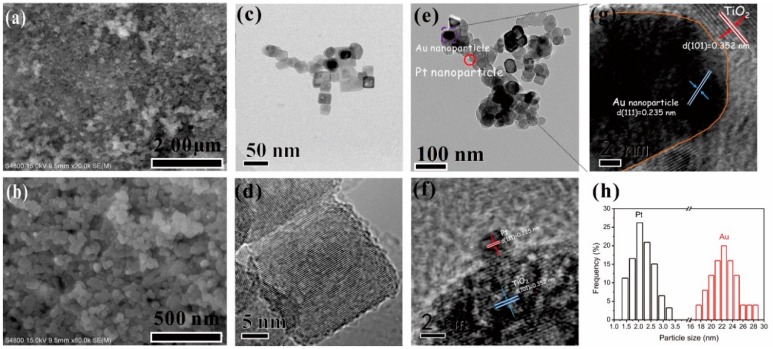
SEM image of Au_2_Pt_2_/N–TiO_2_ (**a**,**b**) and TEM and high-resolution TEM (HRTEM) images of N–TiO_2_ (**c**,**d**) and Au_2_Pt_2_/N–TiO_2_ (**e**–**g**). Nanoparticle size distributions of Au and Pt in Au_2_Pt_2_/N–TiO_2_ (**h**). Scale bar (**e**,**f**): 2 nm.

**Figure 2 nanomaterials-09-00391-f002:**
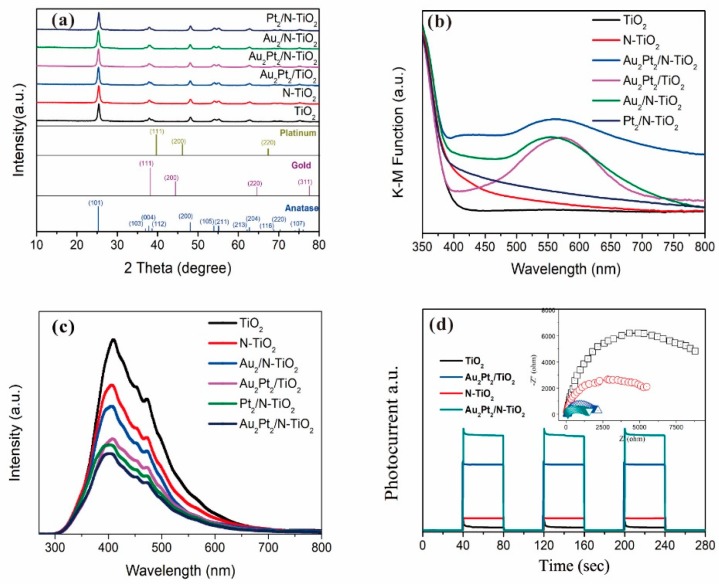
X-ray diffraction (XRD) patterns (**a**), ultraviolet–visible light (UV–Vis) diffuse reflectance spectra (DRS) (**b**), and photoluminescence spectra (**c**) of prepared photocatalysts. Photocurrent transient response (**d**) and electrochemical impedance spectroscopy (EIS) Nyquist plots (inset) of the sample electrodes of TiO_2_, N–TiO_2_, Au_2_Pt_2_/TiO_2_, Au_2_Pt_2_/N–TiO_2_ under visible-light irradiation.

**Figure 3 nanomaterials-09-00391-f003:**
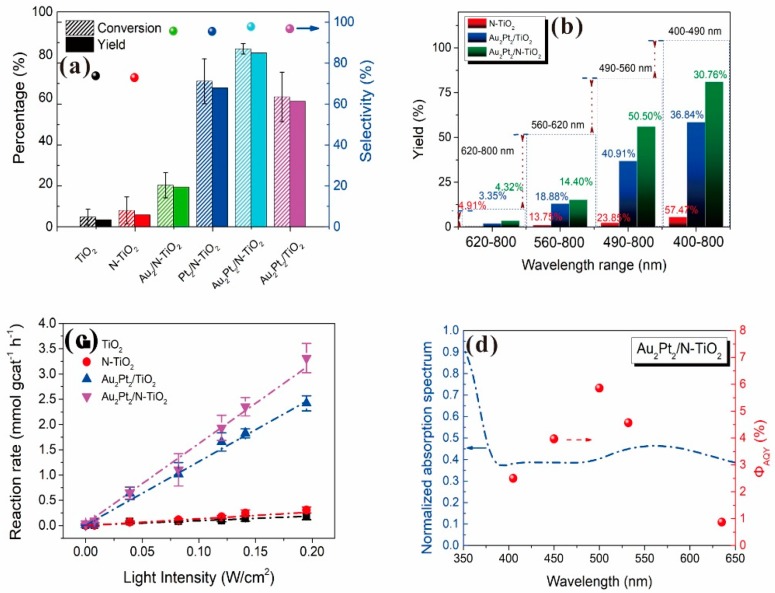
(**a**) Conversion, yield, and selectivity for the photo-oxidation of benzyl alcohol to benzaldehyde over prepared photocatalysts. (**b**) The dependence of yield and irradiation wavelength over photocatalysts for the selective oxidation of benzyl alcohol. (**c**) The rate of photocatalytic reaction over TiO_2_, N–TiO_2_, Au_2_Pt_2_/TiO_2_, and Au_2_Pt_2_/N–TiO_2_ as a function of irradiance intensity. (**d**) Diffuse-reflectance UV–Vis spectra of Au_2_Pt_2_/N–TiO_2_ photocatalyst and the quantum yield for the formation of benzaldehyde under a multiple-wavelength laser light source. The apparent quantum yield was calculated using the equation ΦAQY = (Y_vis_ − Y_dark_)/*N* × 100%, where Y_vis_ and Y_dark_ are the yields of photocatalytic reaction under irradiation or dark conditions, and *N* is the number of incident photons in the reaction vessel.

**Figure 4 nanomaterials-09-00391-f004:**
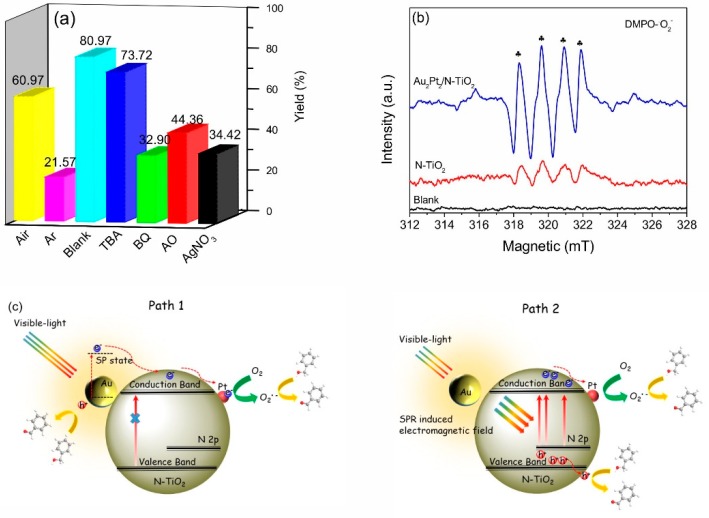
(**a**) Controlled experiment using different radical scavengers; (**b**) ESR spectra collected using 5,5-dimethyl-1-pyrroline-N-oxide ((DMPO-·O_2_^−^)) as the spin trap. (**c**) A plausible mechanism for the photo-oxidation of benzyl alcohol over Au_2_Pt_2_/N–TiO_2_ under visible-light irradiation.

**Table 1 nanomaterials-09-00391-t001:** Photocatalytic selective oxidation of various aromatic alcohols on Au_2_Pt_2_/N–TiO_2_
*^a^*.

Entry	Substrate	Product	Yield *^b^*(%)	Selectivity *^c^*(%)
1	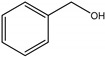	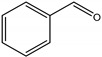	80.9	97.7
2	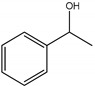	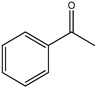	78.1	79.6
3 *^d^*	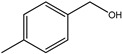	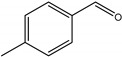	87.9	97.9
4	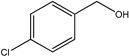	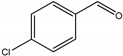	63.8	76.9
5 *^e^*	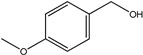	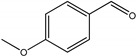	91.2	100
6	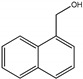	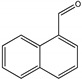	50.1	54.2
7	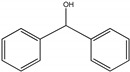	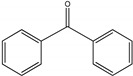	52.2	75.1

*^a^* Solvent: 1.5 mL of trifluorotoluene; substrate: 0.1 mmol aromatic alcohol; photocatalyst: 10 mg of Au_2_Pt_2_/N–TiO_2_; 1 atm of O_2_, reaction temperature was maintained at 30 °C, for 2.5 h of reaction time. *^b^* Yield of aromatic aldehyde; *^c^* selectivity of aldehyde or ketone; *^d^* 1 h of reaction time; *^e^* 1 h of reaction time.

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
