# Peer review of "Coupling Plasmonic and Cocatalyst Nanoparticles on N–TiO2 for Visible-Light-Driven Catalytic Organic Synthesis"

_nanomaterials, 2019, doi:10.3390/nano9030391_

Round 1
Reviewer 1 Report
Review of the manuscript: Coupling plasmonic and cocatalyst nanoparticles on N-TiO2 for visible light-driven catalytic organic synthesis.
By Wang et al.
The manuscript reports the use of AuPt/N-doped TiO2 nanoparticles as photocatalysts for the oxidation of benzyl- and many other aromatic alcohols.
The approach is based on the rational combination of different components: gold nanoparticles promote the injection of hot electrons obtained from visible light excitation of localized surface plasmon resonance, N-doping introduces intraband gap states, and Pt nanoparticles act as electron scavengers, which promotes charge separation and prevents electron-hole recombination. The manuscript is well-written and reports interesting experimental data, which can represent a valuable contribution to the “hot” field of plasmon-enhanced catalysis. In my opinion, it could be accepted by addressing the following points:
Major:
At least two points are not clear about the interpretation of the reaction mechanism:
1) Figure 3c: The dependence of reaction rate on the intensity of irradiating light is linear in all the four samples analysed (TiO2, N-TiO2, Au2Pt2/TiO2, Au2Pt2/-N-TiO2). However, in the case of SPR-mediated catalysis, a superlinear dependence has been often observed (see reference 40). A justification for these differences should be provided.
2) Figure 3d: The trend of quantum yield does not match that of the optical absorption spectrum, which contrasts with the interpretation of the reaction mechanisms. The authors should clarify this point.
Minor:
-Figure 1: The scale bars of SEM images are not clear and should be better displayed.
- Further references on the role of plasmon-mediated production of radical surface species (see for example, Small 2013, 19, 3301, Phys. Chem. Chem. Phys. 2011, 13, 886) and their role in controlling rate and selectivity of the reactions should be added to put the discussion in a proper context.
Author Response
Dear Reviewer:
Thanks for your letter and for the reviewer's comments concerning our manuscript entitled “Coupling plasmonic and cocatalyst nanoparticles on N–TiO2 for Visible-Light-Driven Catalytic Organic Synthesis” Those comments are all valuable and very helpful for revising and improving our paper, as well as the important guiding significance to us researches. We have studied comments carefully and made correction which we hope to meet with approval.
Point 1: Figure 3c: The dependence of reaction rate on the intensity of irradiating light is linear in all the four samples analyzed (TiO2, N-TiO2, Au2Pt2/TiO2, Au2Pt2/N-TiO2). However, in the case of SPR-mediated catalysis, a super-linear dependence has been often observed (see reference 40). A justification for these differences should be provided.
Response 1: As the reviewer suggested that the effect of irradiation intensity can be divided into two parts, which is dominated by the irradiation intensity critical point. It is reported that the correlation between the reaction rate and the low irradiation intensity is linearly correlated (J. Am. Chem. Soc. 2012, 134, 14526−14533 and J. Am. Chem. Soc. 2015, 137, 1956-1966); while the high illumination intensity exhibits a super-linear correlation (Nature Materials. 2012, 11, 1044-1050, Green Chem., 2014, 16, 331–341). Here, in our experiment, the irradiation intensity is not high enough to get the critical point (usually above ∼300 mW cm−2), so there was a linear correlation between the reaction rate and the illumination intensity. Here, we added these published literatures to the revised main text to elucidates this reaction mechanism (line 193-196)
Point 2: The trend of quantum yield does not match that of the optical absorption spectrum, which contrasts with the interpretation of the reaction mechanisms. The authors should clarify this point.
Response 2: Regarding to this reviewer's suggestion, we considered this question carefully. We believed that there are two possible mechanisms may affect the performance of photocatalytic activity, namely as, hot-electron injection and PIRET. Upon irradiation of visible-light, following light absorption and SPR excitation in these nanostructures, electromagnetic decay takes place on a femtosecond timescale non-radiatively by transferring the energy to hot electrons, then these “hot enough” electrons with high energy would inject into the N-TiO2 conduction band. In this manner, the apparent quantum yield would well fit with the pattern of plasmonic metal absorption spectrum. On the other hand, in this case, Nitrogen-doping introduces a new intra-bandgap level above TiO2 valence band, which can resonate with the electromagnetic field generated by the gold SPR effect, and the electromagnetic field is then able to improve the generation of photocarriers from intra-bandgap levels to the TiO2 conduction band through PIRET process. As a result, it will further increase the photocatalytic efficiency. Therefore, the trend of apparent quantum yield doesn’t match well with the trend of the absorption spectrum as reported in previous papers. In the revised manuscript, we added this part of elucidation to further clarify this point (line 180-193).
Point 3: The scale bars of SEM images are not clear and should be better displayed.
Response 3: We are very sorry for our negligence of this detail, we’ve modified scale bars of all SEM and TEM images in our revision article (line 94-95).
Point 4: Further references on the role of plasmon-mediated production of radical surface species (see for example, Small 2013, 19, 3301, Phys. Chem. Chem. Phys. 2011, 13, 886) and their role in controlling rate and selectivity of the reactions should be added to put the discussion in a proper context.
Response 4: As the reviewer suggested, we added the role of plasmon-mediated production of radical surface species in controlling rate and selectivity of the reactions in the revised manuscript (line 231, 247-250).
We appreciate your warm work earnestly and hope that the correction will meet with approval. Once again, thank you very much for your comments and suggestions.
Yours sincerely,
Yannan Wang
Reviewer 2 Report
The manuscript presents the use of Au and Pt co-cocatalysts for the promotion of the photocatalytic activity of N-doped TiO2 in the visible region. The prepared materials were extensively characterized and tested towards the selective oxidation of aromatic alcohols. A possible reaction mechanism is also proposed on the grounds of radical scavenger studies as well as characterizations of the photocatalyst electrochemical properties. In my opinion, the work can be published upon minor improvements.
During photocatalytic tests an irradiation > 460 nm was adopted. The choice of this cut-off wavelength should be better highlighted.
Photocatalytic results should be reported with estimated standard deviations. Photolysis tests should be reported as well.
Significant digits in Table S1 should be checked.
The English text should be revised (e.g., in the abstract “by a by a photo-deposition”, line 157 “which suggest that the overlapped the intrinsic absorption”).
Author Response
Dear Reviewer:
Thanks for your letter and for the reviewer's comments concerning our manuscript entitled “Coupling plasmonic and cocatalyst nanoparticles on N–TiO2 for Visible-Light-Driven Catalytic Organic Synthesis” Those comments are all valuable and very helpful for revising and improving our paper, as well as the important guiding significance to us researches. We have studied comments carefully and made correction which we hope to meet with approval.
Point 1: During photocatalytic tests an irradiation > 460 nm was adopted. The choice of this cut-off wavelength should be better highlighted.
Response 1: As the reviewer suggested, we’ve highlighted light source wavelength in our revised manuscript (line 17).
Point 2: Photocatalytic results should be reported with estimated standard deviations. Photolysis tests should be reported as well.
Response 2: Thanks for your constructive suggestion, we’ve added estimated standard deviations and photolysis on the photocatalytic results in our revised manuscript (line 169-172).
Point 3: Significant digits in Table S1 should be checked.
Response 3: We are very sorry for our negligence of this detail, we’ve modified the caption of Table S1.
Point 4: The English text should be revised (e.g., in the abstract “by a by a photo-deposition”, line 157 “which suggest that the overlapped the intrinsic absorption”).
Response 4: We are very sorry for our negligence of this detail, we’ve revised these sentences in our revised manuscript (line 14, 166).
We appreciate your warm work earnestly and hope that the correction will meet with approval. Once again, thank you very much for your comments and suggestions.
Yours sincerely,
Yannan Wang

Reviewer 3 Report
The article presented by Wand and co-workers describes the synthesis and addition of bimetallic nanoparticles onto titania supports and the influence of doping the latter with nitrogen. This piece of work is worth publishing, the topic is interesting and the results look promising given the photocatalytic response in the visible range. Still, the manuscript needs some ammendments before final acceptance.
GENERAL COMMENTS
- The work requires a moderate to exhaustive English revision.
- The legends and scale bars in Figures are ilegible. All figures require modification in this sense in order to make the figures properly readable.
SPECIFIC QUERIES
- Additional TEM images with proper magnification need to be included. It remains unclear how Au is distributed. Can the authors comment on any potential galvanic replacement observed between Au and Pt (see for instance Nanoscale, 2015, 7, 10152–10161)?
- Additional STEM-EDX analysis would be desirable to evaluate the distribution of both metal species and if any bimetallic phases are formed (before and after reaction).
- The photodeposition methodology must be explained in detail in order to make it reproducible for any author. Please expand and complete properly for a revised version of this manuscript.
- Please include the complete chemical name and purchase company for [Bmimi][BF4]-
- Please check Figure S1. The legend does not correspond to the image shown.
Author Response
Dear Reviewer:
Thanks for your letter and for the reviewer's comments concerning our manuscript entitled “Coupling plasmonic and cocatalyst nanoparticles on N–TiO2 for Visible-Light-Driven Catalytic Organic Synthesis” Those comments are all valuable and very helpful for revising and improving our paper, as well as the important guiding significance to us researches. We have studied comments carefully and made correction which we hope to meet with approval.
Point 1: The work requires a moderate to exhaustive English revision.
Response 1: As all reviewers mentioned, there are some obvious errors in the English expression of this work. We are very sorry for our negligence of this detail, we’ve revised these sentences in our revised manuscript.
Point 2: The legends and scale bars in Figures are illegible. All figures require modification in this sense in order to make the figures properly readable.
Response 2: We are very sorry for our negligence of this detail, we’ve modified scale bars of all SEM and TEM images in our revision article and revised captions of all figures. (line 93-95, 150-153, 209-217, 256-258)
Point 3: Additional TEM images with proper magnification need to be included. It remains unclear how Au is distributed. Can the authors comment on any potential galvanic replacement observed between Au and Pt (see for instance Nanoscale, 2015, 7, 10152–10161)?
Response 3: As the reviewer suggested, we’ve uploaded more TEM images with different magnifications in figure S1 of Supplementary material. As shown in the attached TEM images, gold nanoparticles with an average size of ~30 nm can be clearly observed instead of the presence of bimetallic Au-Pt alloy. According to the observation by Uson et al, it is suggested that the size of the metal nanoparticles is critical for modifying the chemical composition of the resulting nanomaterials, and the bimetallic alloy usually can be observed in the case of small gold (~7 nm) and platinum (~2 nm) nanoparticles. Here, in this case, the gold nanoparticles size is larger than 20 nm, the formation of the bimetallic alloy NPs is not thermodynamically favored, and the segregation of gold and platinum nanoparticles is maintained. In previous reports, platinum nanoparticle, as the cocatalyst, can form a Schottky barrier at the interface of semiconductor, serving as the “electron trapper” to improve migration and separation of photo-charges, which plays an important role in photocatalytic reactions. Considering the reviewer's suggestion, we’ve added description in the revised manuscript (line 97-102).
Point 4: Additional STEM-EDX analysis would be desirable to evaluate the distribution of both metal species and if any bimetallic phases are formed (before and after reaction).
Response 4: Thanks for your constructive suggestion, we’ve performed TEM and EDX analysis of photocatalyst after reaction (Figure S6 in the Supplementary material), and it can be observed that the segregation of gold and platinum nanoparticles is still maintained. We believe this is because the metal nanoparticles are supported by the TiO2 substrate material, which limits the diffusion of metal atoms.
Point 5: The photodeposition methodology must be explained in detail in order to make it reproducible for any author. Please expand and complete properly for a revised version of this manuscript.
Response 5: As reviewer suggested that we’ve expanded the photodeposition methodology in our revised manuscript (in Supplementary material).
Point 6: Please include the complete chemical name and purchase company for [Bmim][BF4]
Response 6: We are very sorry for our negligence of this detail, we’ve the complete chemical name of [Bmim][BF4] (in Supplementary material).
Point 7: Please check Figure S1. The legend does not correspond to the image shown.
Response 7: Thanks for your suggestion, we are very sorry for our negligence of this detail, we’ve revised captions of all figures (in Supplementary material).
We appreciate your warm work earnestly and hope that the correction will meet with approval. Once again, thank you very much for your comments and suggestions.
Yours sincerely,
Yannan Wang

Round 2
Reviewer 1 Report
The manuscript has been improved and can ba accepted in the present form